# Whole-Genome *k*-mer Topic Modeling Associates Bacterial Families

**DOI:** 10.3390/genes11020197

**Published:** 2020-02-14

**Authors:** Ernesto Borrayo, Isaias May-Canche, Omar Paredes, J. Alejandro Morales, Rebeca Romo-Vázquez, Hugo Vélez-Pérez

**Affiliations:** 1Electronics Department, CUCEI, Universidad de Guadalajara, Jalisco 44100, Mexico; ernesto.borrayo@academicos.udg.mx; 2Computer Sciences Department, CUCEI, Universidad de Guadalajara, Jalisco 44100, Mexico; imay@itchetumal.edu.mx (I.M.-C.); omar.paredes@academicos.udg.mx (O.P.); jalejandro.morales@academicos.udg.mx (J.A.M.); rebeca.romo@academicos.udg.mx (R.R.-V.); 3Instituto Tecnológico de Chetumal, Quintana Roo 77000, Mexico

**Keywords:** topic model, bacteria genome comparison, alignment-free

## Abstract

Alignment-free *k*-mer-based algorithms in whole genome sequence comparisons remain an ongoing challenge. Here, we explore the possibility to use Topic Modeling for organism whole-genome comparisons. We analyzed 30 complete genomes from three bacterial families by topic modeling. For this, each genome was considered as a document and 13-mer nucleotide representations as words. Latent Dirichlet allocation was used as the probabilistic modeling of the corpus. We where able to identify the topic distribution among analyzed genomes, which is highly consistent with traditional hierarchical classification. It is possible that topic modeling may be applied to establish relationships between genome’s composition and biological phenomena.

## 1. Introduction

Alignment-free sequence algorithms have been widely explored for sequence analyses due to their ability to render relatively accurate results while lowering algorithm complexity [1]. Among them, many utilize segments of genomic sequences of length *k*. Such segments are symbolic representations of a four-letter alphabet called words or *k*-mers. *K*-mer frequency content comparisons have proven high-accuracy [2,3]. Moreover, Fofanov et al. [4] described that bacterial genome word distributions for various *k*-mer lengths are not random, which suggests that genome *k*-mer distribution differs between families. And while Zhang et al. [5] demonstrated how virus whole-genome word distributions clusters according to their taxonomy, the *k*-mer alignment-free comparisons for whole-genome sequence analysis still remains a challenge.

Methods based on word count or *k*-mer frequency can be summarized into: *(i) composition vectors*, where genomic content is represented in matched/mismatched occurrence-vector motifs to classify studied sequences either by pairwise-distance measures or machine-learning approaches [6,7]; *(ii) information theory*, which include a set of methodologies that evaluate the shared Shannon-view informational content (i.e., mutual information and complexity/data compression) among the sequences studied [3,8]; *(iii) motif composition*, where probabilistic methods model genomic sequences and compare the motifs’ expected frequencies rendered by their respective models [4]; and *(iv) D_2_ statistics*, that describe the compared sequences as the D_2_ model based on the shared words’ statistics [9,10].

A general practice in these methods is to perform pairwise comparisons, which renders a version of a distance matrix that is typically solved by maximum parsimony, neighbor joining or other tree-building methods. Alternate solutions for a distance matrix include clustering algorithms, (such as *k*-means) [11,12,13] and probabilistic topic modeling. Also, machine-learning methods –such as Support Vector Machines–, have been used for sequence classification [14].

Altogether, these methods have been implemented into different biological applications, from small viral sequences analysis [15] to complete genome analysis [3], with a considerably increase in computational requirements as the analyzed sequences number and size increases [1] even when highly optimized algorithms are applied [16,17]. Also, most of these processes involve prior knowledge of the nature of the sequences [18], classifier training [19], and they do not necessarily provide insight into the genomic composition of the evaluated sequences nor regarding other significant biological phenomena.

Topic modeling is a suite of algorithms aimed to discover certain lexicon-themed structures in a corpus of several documents [20] based on their word-list distribution. Some applications outside language natural processing have been reported in the literature: drug classification according to safety and therapeutic use [21], image clustering [22], audio [23] and music [24] analysis, and more recently, the *k*-mer sequence analysis [25].

Most of the topic modeling sequence analysis have consisted in single gene-based non-overlapping word analysis and lexicon clustering of such genes [25,26]. To our knowledge, no work has used whole-genome word corpus –bacterial or otherwise– clustered by topic modeling in organism genomic comparisons. In this paper, we explore a proof-of-concept cluster topic modeling of three bacterial families based on their genome word-list distribution.

## 2. Methods

### 2.1. Corpus & Bacterial Families

To apply the cluster topic modeling to sequence analysis and genome comparison, we selected and downloaded 30 complete genomes from the NCBI database (https://www.ncbi.nlm.nih.gov/, June 2019), and treated each genome as a document. Ten genomes were downloaded for three pathogenic bacterial families: *Chlamydiales*, *Vibrionaceae*, and *Yersiniaceae* respectively (Accession numbers and species details are referenced in Table 1). Bacterial families were chosen according to three criteria: clear biological differences among them, enough complete genomes in the family for the analysis, as well as in numbers that allowed computational manageability. Complete genomes were selected over incomplete genomes/scaffolds since the former are annotated and curated, and therefore their information has a very low variation rate between versions. The algorithm can be applied to incomplete genomes.

Given that the corpus is the full collection of words that are putatively present in a group of documents and, that the subcorpus is a collection of unique words that composes a particular document, we studied both the corpus and subcorpus of the genomes through the presence or absence of particular words on each genome. We considered the word –or *k*-mer– to be an overlapped genome’s *k*-size sub-string. Therefore we can establish that in any given length *l* genome, the total number of *k*-mers (*N*) is equal to l-k+1. The *k*-mer size selection was determined by the Cumulative Relative Entropy metric [8], which is a second-order Markov estimator that reflects the information gain of a word of size *k*. As the value approaches zero, the accuracy to estimate longer features trades-off with computer-time geometric increase.

### 2.2. Topic Model

Probabilistic topic modeling refers to a suite of algorithms that are assembled in order to discover, classify and annotate thematic information in large documents (Figure 1). The principal advantage of these algorithms is that they do not require prior document information –such as previous annotations or labeling– as the topics emerge from the original texts’ analysis [27]. Topic models’ analysis is build up on the concept that documents can be considered to be as mixtures of topics, where a topic is generated by the probability distribution of words [28]. Therefore, it is possible to extract the most recurrent themes –or topics– shared by a corpus of sequences [25] which in this work, is the corpus composed by the selected genomes. In order to do so, a series of *N* words can represent a document *d*: $d=(w_{1},w_{2},\ldots ,w_{N})$. The generative model for documents can be expressed by the following probability distribution:
(1)$$P\left(w_{i}\right)=\sum_{j=1}^{T}P\left(w_{i}|z=z_{j}\right)P\left(z=z_{j}\right)$$
where $P(w_{i})$ is the probability of the word $w_{i}$ in a given document; $P(z=z_{j})$ is the probability of choosing a word from a topic $z_{j}$ for the current document; $P(w_{i}|z=z_{j})$ is the probability of sampling the word $w_{i}$, given the topic $z_{j}$; and *T* is the number of topics [25,29]. In this context, a corpus is defined as a collection of *M* documents denoted by $D=w_{1},w2,\ldots ,w_{M}$ [30].

The probabilistic topic model used in this work is the Latent Dirichlet Allocation (LDA). The main idea is that documents are represented as random mixtures over latent topics, where each topic is characterized by a distribution over words. LDA assumes the following generative process for each document *w* in a corpus *D* [30]:
Choose N∼ Poisson (ξ).Choose θ∼ Dir (α).For each of the *N* words $w_{n}$:
Choose a topic $z_{n}$∼ Multinomial (θ).Choose a word $w_{n}$ from $p(w_{n}|z_{n},\beta )$, a multinomial probability conditioned on the topic $z_{n}$.



In this work, we used the *topicmodels* R package [25].

## 3. Results and Discussion

Optimal *k* was determined as 13-mer by Cumulative Relative Entropy metric, since it rendered values under the suggested threshold (Figure 2) without a considerably trade-off with computational requirements.

From the $4^{13}$ possible 13-mers, only about 8% (41′392,339) where significantly present in the 30 complete genomes included in this study. Thus, we reduced our corpus to those *k*-mer words that where present in a number of genomes equal or greater than ten –the number of species on each evaluated family–. This provided with a discrimination criteria that allowed us to select the putative useful *k*-mers for subcorpus classification. The selected corpus consisted in 211,680 13-mers that where found 2′419,034 overall in the evaluated genomes.

Once the corpus was established, we applied the LDA algorithm to model topics that could differentiate the genomes based on their taxonomy. The LDA was compiled with Gibbs sampling and the default parameters for 3 topics. The main result of the algorithm is the probability distribution for each genome to all topics based on its subcorpus.

Given that topics are sets of highly-probable *k*-mers occurring in each genome, we can infer that those lexicons carry out similar biological functions in the evolutionary process. Therefore, closely functional genomes aggregate similar *k*-mers. Our results (Figure 3) show that each cluster is highly represented by a topic: topic 1 for *Chlamydiales*, topic 2 for *Vibrionaceae*, and topic 3 for *Yersiniaceae*. This suggests that these genomes tend to adopt a homogeneous lexicon that agrees with a selective process and outline the biological functions or traits that could fulfilled.

Figure 3 shows the hierarchical clustering of the probability distributions of each genome for the three topics. The three well-defined clusters correspond with the bacterial families used in this work. Each cluster tends to be more related with a single topic. It is of note the case of the *Yersineaceae* family that is represented by topic 3 and is clearly divided in two subclusters, correlating with the two distinct genera that these genomes belong within this family. The difference between both genera is their probability of sharing words with topic 2.

The clustering method groups together most of the genomes based on their families. One exception is *Serratia symbiotica strain STs*, that shows a high-probability for words in topic 1, and thus is clustered together with the *Chlamydiales* family. Although we cannot relate the rendered topic-based organism classification with particular intrinsic *corpus* characteristics, the clustering presents an interesting result. *S. symbiotica* is not allocated as expected with its traditional taxonomical family *Yersiniaceae*, but with *Chlamydiales* instead. This may be explained by the fact that organisms with high presence of topic 1 are conventionally classified as obligate intracellular parasites, while the rest are either facultative intracellular or extracellular parasites. As among *Yersiniaceae S. symbiotica* is an obligate intracellular symbiont in Aphids [31], it is possible that either topic 1 *corpus* lexicon is related to specific mechanisms for intracellular biological relationship while topic 2 and 3 lexicons may establish a correspondence with mechanisms related to free-life capabilities. This explanation needs to be taken with reserve, mainly by the fact that genome sizes are considerably smaller in those obligate intracellular parasites and even more in the symbiont. One of the multiple factors that could be affecting *k*-mer topic identity assignation by the algorithm is the genome size. *k*-mer sequences and topic’s pertinence’s probability will shift more drastically in a smaller genome. However, this result suggests that there is a high possibility that in fact topic 1 narrative is related to intracellular mechanisms and invites to further research.

*k*-mer analysis has become a popular approach for sequence comparison [32,33]. So far, there are three pivotal previous studies that classified complete genomes by this approach. Two of them focused on viral genomes [5,34], while the remaining one on the classification of bacterial genomes [3]. Complete genomes comparison is still challenging when under- and over-represented words are considered, and becomes even more so as the genome size increases. Sims and Kim [3] removed sets of words terms in order to classify bacterial genomes from different genera. With topic modeling we were able to include over-represented words with an accurate bacterial family’s classification (Figure 4), which could be explained by the fact that the removed lexicon may carry meaningful biological information. A plausible solution for this methodological artifact is the incorporation of natural language processing –such as LDA– which has proven success at single gene comparison level [25] and that takes into consideration the over-represented terms. Our results show that this rationale is efficient in the bacterial family’s classification at a whole genome scale comparison.

Most topic modeling algorithms have been successfully applied to single gene sequences [25,35], however, with the exponential growth of whole-genome data, topic modeling can be implemented to span complete genome analysis. Our approach, is based on the concept that genomic lexicon may be fixed as a representation of biological processes in organisms and therefore used as discriminators between them. The expected outcome is *k*-mers being clustered according to topics with putative similar lexicons. As organisms are clustered based on the frequency of *k*-mer’s probability to each topic, it is possible to hypothesise that each topic is part of a lexicon related to a group of similar biological processes or functions. Backenroth et al. [36] demonstrated that it is possible to predict how regulatory-sequence changes diverge in topics to predict tissue-specific functional effects. Functional word analysis may be implemented as an extension of topic attributes, becoming another approach for future work that involves supervised topic modeling.

Topic modeling techniques have been previously used in bioinformatics to classify sequences ether according to their coding genes [26], 16S rDNA [25], or other biological activities [37,38,39,40]. La Rosa and colleagues compared LDA vs. support vector machine to classify bacterial families. They concluded that while both methods are precise for full-length 16S rDNA sequences, only LDA is robust enough for smaller *k*-mers. The explanation for this lies in the capacity for supervised machine learning methods to generalize information from their training sets via previously selected features. In contrast, LDA is an unsupervised machine learning method able to correctly classify a small-*k*-mer corpus composed of complete genomes. This implementation could be extrapolated to incomplete genomes (e.g., when analyzing bacterial high-throughput sequencing) by considering that word samples do not necessarily mean a corpus and biological significance interpretation can be compromised.

## 4. Conclusions

Herein we establish that the Topic model can be applied to complete genome comparison with results that are consistent with the current bacterial taxonomy. The topic modeling has the advantage of not needing a selection of characteristics to differentiate genomes according to their taxonomy. It is possible that future exploration will help to establish relationships between genome’s composition and other significant biological phenomena.

## Figures and Tables

**Figure 1 genes-11-00197-f001:**
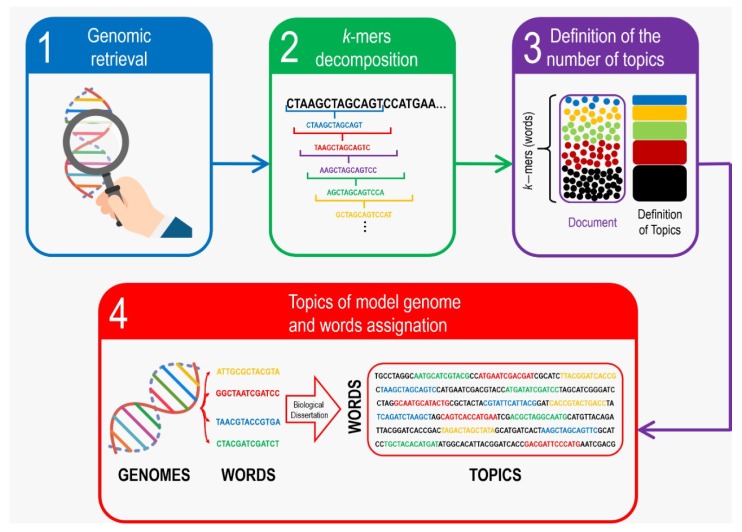
Schematic procedure of Whole-genome *k*-mer topic modeling association. To-be-compared genomes are retrieved either from databases or from experimental procedures (1) to be decomposed into *k*-mers (2) and then analyzed in order to determine the adequate topic number (3) to finally perform the topic classification as summarized in (4).

**Figure 2 genes-11-00197-f002:**
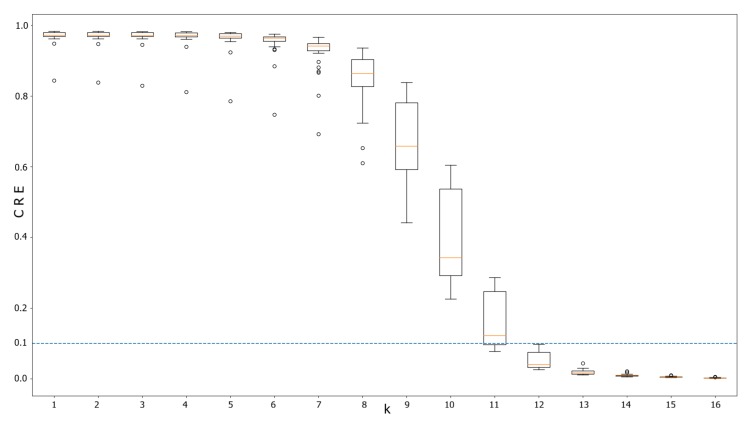
Box plot for Cumulative Relative Entropy for different *k* sizes involving 30 bacterial genomes. The suggested threshold is below 0.1 to maximize differences between the genomes. Notice that *k* = 13 is the first *k* where neither value is above the threshold.

**Figure 3 genes-11-00197-f003:**
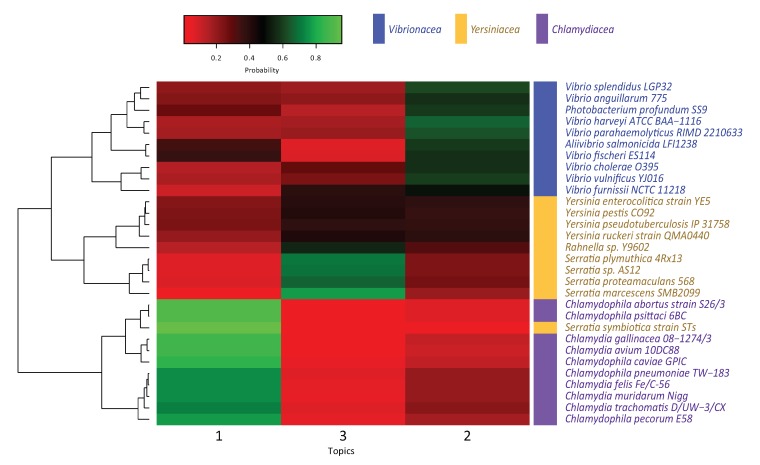
Phylogenomic classification of bacterial families *Chlamydiales*, *Vibrionaceae*, and *Yersiniaceae* based on topic modeling (In this work, three topics).

**Figure 4 genes-11-00197-f004:**
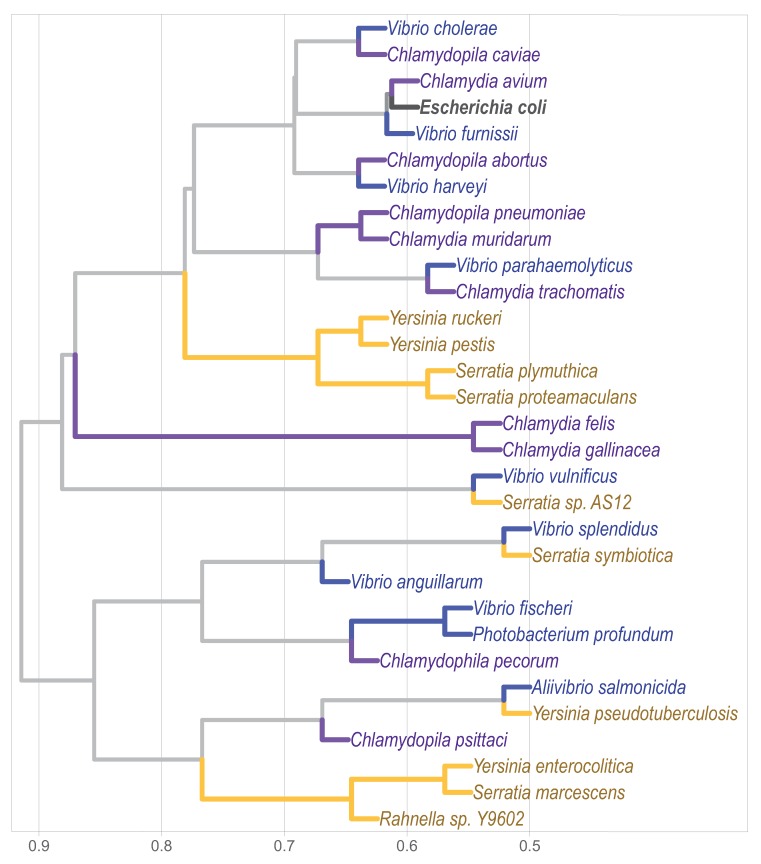
Phylogenomic classification of bacterial families *Chlamydiales*, *Vibrionaceae*, and *Yersiniaceae* based on the methodology of Sims and Kim [3], and including over-represented words.

**Table 1 genes-11-00197-t001:** Genomic information from bacteria selected for Whole-genome *k*-mer topic modeling association.

Accession No.	Family	Organism	Genome Size (bp)
AE001273.1		*Chlamydia trachomatis D/UW-3/CX*	1,042,519
AE002160.2		*Chlamydia muridarum Nigg*	1,072,950
AE009440.1		*Chlamydophila pneumoniae TW-183*	1,225,935
AE015925.1		*Chlamydophila caviae GPIC*	1,173,390
AP006861.1	*Chlamydiales*	*Chlamydia felis Fe/C-56*	1,166,239
CP002549.1		*Chlamydophila psittaci 6BC*	1,171,660
CP002608.1		*Chlamydophila pecorum E58*	1,106,197
CP006571.1		*Chlamydia avium 10DC88*	1,041,170
CP015840.1		*Chlamydia gallinacea 08-1274/3*	1,059,583
CR848038.1		*Chlamydophila abortus strain S26/3*	1,144,377
BA000031.2		*Vibrio parahaemolyticus RIMD 2210633*	3,288,558
BA000037.2		*Vibrio vulnificus YJ016*	3,354,505
CP000020.2		*Vibrio fischeri ES114*	2,897,536
CP000626.1		*Vibrio cholerae O395*	1,108,250
CP000789.1	*Vibrionaceae*	*Vibrio harveyi ATCC BAA-1116*	3,765,351
CP002284.1		*Vibrio anguillarum 775*	3,063,912
CP002377.1		*Vibrio furnissii NCTC 11218*	3,294,546
CR354531.1		*Photobacterium profundum SS9*	4,085,304
FM178379.1		*Aliivibrio salmonicida LFI1238*	3,325,165
FM954972.2		*Vibrio splendidus LGP32*	3,299,303
AL590842.1		*Yersinia pestis CO92*	4,653,728
CP000720.1		*Yersinia pseudotuberculosis IP 31758*	4,723,306
CP000826.1		*Serratia proteamaculans 568*	5,448,853
CP002505.1		*Rahnella sp. Y9602*	4,864,217
CP002774.1	*Yersiniaceae*	*Serratia sp. AS12*	5,443,009
CP006250.1		*Serratia plymuthica 4Rx13*	5,328,010
CP016940.1		*Yersinia enterocolitica strain YE5*	4,593,248
CP017236.1		*Yersinia ruckeri strain QMA0440 isolate 14/0165-5k*	3,856,634
HG738868.1		*Serratia marcescens SMB2099*	5,123,091
LN890288.1		*Serratia symbiotica strain STs*	650,317

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
