# Peer review of "Whole-Genome k-mer Topic Modeling Associates Bacterial Families"

_genes, 2020, doi:10.3390/genes11020197_

Round 1

Reviewer 1 Report

The authors extended the application of k-mer topic modelling from genes to genomes and tried to establish relationships between bacterial genomes. However, the work shown in their manuscript is very preliminary, with no justification of k-mer size selection, with no comparison of classification being demonstrated, with neither advantages nor disadvantages being discussed. Besides, only a limited number of complete genomes were used, without incomplete genomes included in test, without genome selection justification. A significant improvement is needed to demonstrate the advantages of the method/application.

Minors include:

1) Methods could be simplified since algorithm of reference 14 is mainly used;

2) Precision: e.g. Line 33 "a few" should be "three";

3) Citation after names: e.g. Line 15 "Fofanov et al." should be "Fofanov et al. [4]" ;

4) Figure legend and Table description should be more detailed.

Author Response

Manuscript ID: genes-701637

TitleWhole-genome k-mer topic modelling associates bacterial families

Authors: Ernesto Borrayo, Isaias May-Canche, Omar Paredes, J. Alejandro Morales, Rebeca Romo-Vázquez, & Hugo Vélez-Perez

Reviewer 1:

Comment:

However, the work shown in their manuscript is very preliminary, with no justification of k-mer size selection, with no comparison of classification being demonstrated, with neither advantages nor disadvantages being discussed. Besides, only a limited number of complete genomes were used, without incomplete genomes included in test, without genome selection justification.  A significant improvement is needed to demonstrate the advantages of the method/application.

Response.

We agree that this is a preliminary work. It is also clear that there is room for improvement thanks to your accurate suggestions. We mention the theoretical support for k-mer size selection in the methodology section “The k-mer size selection was determined by the Cumulative Relative Entropy metric [17]”. Figure i (please find  the figure in the attached file) shows the result for all genomes. We would not like to bias the reader’s attention from the main topic, which is why we agreed not to include the following plot.

[17] Sims, G.E.; Jun, S.R.; Wu, G.A.; Kim, S.H. Alignment-free genome comparison with feature frequency profiles (FFP) and optimal resolutions. Proceedings of the National Academy of Sciences 2009, 106, 2677–2682.

Figure i. Boxplot for Cumulative Relative Entropy for the different k sizes involving 30 bacterial genomes. The suggested threshold below 0.1 to maximize differences between the genomes. Notice that k=13 is the first neither value is above the threshold.

The family selection was determined by including families that have clear biological differences among them, that had enough complete genomes for the analysis and in numbers that allowed computational manageability. Complete genomes where selected over incomplete genomes/scaffolds since the formers are annotated and curated, and therefore their information has very low variation rate between versions. The inclusion of several bacterial families on complete genomes and scaffolding is ongoing research in our group.

We also added some new elements to the discussion, with the objective to clarify the aspects that seemed to be insufficient to establish our argument, particularly, regarding its advantages and disadvantages with respect to other methodologies “k-mer analysis has become a popular approach for sequence comparison [23,24].  So far, there are three pivotal previous studies that classified complete genomes by this approach. Two of them focused on viral genomes [5,25], while the remaining one on the classification of bacterial genomes [3]. Complete genomes comparison is still a challenge when the under- and over-represented terms are considered and becomes even more challenging as the size of the analyzed genomes increases. Sims and Kim [3] removed such terms in order to classify bacterial genomes from different genera. The implementation for such strategy with our dataset rendered in a not accurate bacterial family’s classification (Figure 3), which could be explained by the fact that the removed lexicon may carry meaningful biological information. A plausible solution for this methodological artifact is the incorporation of natural language processing –such as LDA– which has proven success at single gene comparison level [15] and that takes into consideration the over-represented terms. Our results show that this rationale is efficient in the bacterial family’s classification at a whole genome scale comparison. (…) Topic modelling techniques have been previously used in bioinformatics to classify sequences ether according to their coding genes [16] or their 16S rDNA [15].  In the latter, La Rosa and colleagues compared LDA vs. support vector machine to classify bacterial families. They concluded that while both methods are precise for full-length 16S rDNA sequences, only LDA is robust enough for smaller k-mers.  The explanation for this lies in the capacity for supervised machine learning methods to generalize information from their training sets via previously selected features. In contrast, LDA is an unsupervised machine learning method able to correctly classify a small-k-mer corpus composed of complete genomes.” And modified the Conclusion accordingly. “Herein we establish that the Topic model can be applied to complete genome comparison with results that are consistent with the current bacterial taxonomy. The topic modelling has the advantage of not needing a selection of characteristics to differentiate genomes according to their taxonomy. It is possible that future exploration by this concept help to establish relationships between genome’s composition and other significant biological phenomena.”.

Comment.

Methods could be simplified since algorithm of reference 14 is mainly used; Precision: e.g. Line 33 "a few" should be "three"; Citation after names: e.g. Line 15 "Fofanov et al." should be "Fofanov et al. [4]"; Figure legend and Table description should be more detailed.

Response.

All minor punctual recommendations have been addressed. After conferring, we determined that “Methods” section is adequate as it is, even though is highly based on LaRosa’s work.

Reviewer 2 Report

In this study, the authors have explored the possibility to use Topic Modeling for organism whole-genome comparison. Basically they have used 13 mer nucleotide to represent as word. This article has several issues. Therefore, I do not recommend it for the publication.

1. How did they end up with 13 mer without providing any systematic analysis.
2. Generally, Kmers have been widely applied in ML-based sequence function prediction. I am wondering how come this analysis contribute the field?

Author Response

Manuscript ID: genes-701637

TitleWhole-genome k-mer topic modelling associates bacterial families

Authors: Ernesto Borrayo, Isaias May-Canche, Omar Paredes, J. Alejandro Morales, Rebeca Romo-Vázquez, & Hugo Vélez-Perez

Review 2:

Comment.

How did they end up with 13 mer without providing any systematic analysis?

Response.

We have included the theoretical support for k-mer size selection in the methodology section “The k-mer size selection was determined by the Cumulative Relative Entropy metric [17]”. We would not like to bias the reader’s attention from the main topic, reason that we agreed not to include the following plot (please find  the figure in the attached file), as it seems unnecessary.

[17] Sims, G.E.; Jun, S.R.; Wu, G.A.; Kim, S.H. Alignment-free genome comparison with feature frequency profiles (FFP) and optimal resolutions. Proceedings of the National Academy of Sciences 2009, 106, 2677–2682.

Figure i. Boxplot for Cumulative Relative Entropy for the different k sizes involving 30 bacterial genomes. The suggested threshold below 0.1 to maximize differences between the genomes. Notice that k=13 is the first neither value is above the threshold.

Comment.

Generally, Kmers have been widely applied in ML-based sequence function prediction. I am wondering how come this analysis contribute the field?

Response.

A more detailed discussion of how the proposed method is related to the state of the art in ML in genomic analysis, has been included. “Topic modelling techniques have been previously used in bioinformatics to classify sequences ether according to their coding genes [16] or their 16S rDNA [15]. In the latter, La Rosa and colleagues compared LDA vs. support vector machine to classify bacterial families. They concluded that while both methods are precise for full-length 16S rDNA sequences, only LDA is robust enough for smaller k-mers.  The explanation for this lies in the capacity for supervised machine learning methods to generalize information from their training sets via previously selected features. In contrast, LDA is an unsupervised machine learning method able to correctly classify a small-k-mer corpus composed of complete genomes.”

Round 2

Reviewer 1 Report

1) Justification of k-mer size selection is critical as both reviewers pointed out. The attached figure should be included in the manuscript. As 13mer is acceptable, my question is: will 14mer, 15mer or others be better? in what price? No test done or discussed.

2) A new method should be applicable, mostly to incomplete genomes. Despite recent NGS advances, the number of complete genomes is low. Besides, classification of complete genomes is relatively easy. Therefore, working on incomplete genomes with taxonomy classification available is more important, at least to my own opinion. Regrettably, no test is done.

3) There are more kmer- and weighted/scored kmer-based whole-genome analysis methods available. Regrettably, they are not reviewed in introduction or discussion. Comparing to these methods with demonstrated advantages and disadvantages is more critical to readers. Regrettably, no comparison is done.

Author Response

Manuscript ID: genes-701637

Title: Whole-genome k-mer topic modelling associates bacterial families

Authors: Ernesto Borrayo, Isaias May-Canche, Omar Paredes, J. Alejandro Morales, Rebeca Romo-Vázquez, & Hugo Vélez-Perez

Reviewer 1:

Comment: 1) Justification of k-mer size selection is critical as both reviewers pointed out. The attached figure should be included in the manuscript. As 13mer is acceptable, my question is: will 14mer, 15mer or others be better? in what price? No test done or discussed.

k-mer justification has been added to both methodology and results as follows:

In methodology:

“The k-mer size selection was determined by the Cumulative Relative Entropy metric [8], which is a second-order Markov estimator that reflects the information gain of a word of size k. As the value approaches zero, the accuracy to estimate longer features trades-off with computer-time geometric increase.”

In Results and Discussion:

 “Optimal k was determined as 13-mer by Cumulative Relative Entropy metric, as with that k it rendered values under the suggested threshold (Figure 2) without a considerably trade-off with computational requirements.”

Figure Inclusion.

Comment:  2) A new method should be applicable, mostly to incomplete genomes. Despite recent NGS advances, the number of complete genomes is low. Besides, classification of complete genomes is relatively easy. Therefore, working on incomplete genomes with taxonomy classification available is more important, at least to my own opinion. Regrettably, no test is done.

An argument regarding the plausibility to implement the method to incomplete genomes has been added to Methods:

“Bacterial families were chosen according to three criteria: clear biological differences among them, enough complete genomes in the family for the analysis, as well as in numbers that allowed computational manageability.  Complete genomes were selected over incomplete genomes/scaffolds since the former are annotated and curated, and therefore their information has a very low variation rate between versions. The algorithm can be applied to incomplete genomes.”;

and also to the Results/Discussion:

“This implementation could be extrapolated to incomplete genomes (e.g. when analyzing bacterial high-throughput sequencing) by considering that word samples do not necessarily mean a corpus and biological significance interpretation can be compromised.”

Comment: 3) There are more kmer- and weighted/scored kmer-based whole-genome analysis methods available. Regrettably, they are not reviewed in introduction or discussion. Comparing to these methods with demonstrated advantages and disadvantages is more critical to readers. Regrettably, no comparison is done.

We added a brief methodological comparison on the Introduction:

“Methods based on word count or k-mer frequency can be summarized into: (i) composition vectors, where genomic content is represented in matched/mismatched occurrence-vector motifs to classify studied sequences either by pairwise-distance measures or machine-learning approaches [6,7]; (ii) information theory,  which include a  set of methodologies that evaluate the shared  Shannon-view informational content  (i.e. mutual information and complexity/data compression)  among the sequences studied [3,8]; (iii) motif composition, where probabilistic methods model genomic sequences and compare the motifs’ expected frequencies rendered by their respective models [4]; and (iv) D2 statistics, that describe the compared sequences as the D2 model based on the shared words’ statistics [9,10].” … “Altogether, these methods have been implemented into different biological applications, from small viral sequences analysis [15] to complete genome analysis [3], with a considerably increase in computational requirements as the analyzed sequences number and size increases [1] even when highly optimized algorithms are applied [16,17]. Also, most of these processes involve prior knowledge of the nature of the sequences [18], classifier training [19], and they do not necessarily provide insight into the genomic composition of the evaluated sequences nor regarding other significant biological phenomena.”

And into the Results/Discussion:

“Complete genomes comparison is still challenging when under- and over-represented words are considered, and becomes even more so as the genome size increases.  Sims and Kim [3] removed such terms in order to classify bacterial genomes from different genera.  With topic modeling we were able to include over-represented words with an accurate bacterial family’s classification (Figure 4), ...”

Reviewer 2 Report

The authors have addressed most of my comments. Still a minor revision is needed.

Kmers have been widely applied in ML-based sequence-function prediction. Here, I have provided latest references (PMID: 31994694, 31099381, 31157855 and 31542696). It would be appropriate to include it in the following sentence
"Topic modelling techniques have been previously used in bioinformatics to classify sequences ether according to their coding"

Author Response

Manuscript ID: genes-701637
Title: Whole-genome k-mer topic modelling associates bacterial families
Authors: Ernesto Borrayo, Isaias May-Canche, Omar Paredes, J. Alejandro Morales, Rebeca Romo-Vázquez, & Hugo Vélez-Perez

Review 2:

Comment.

The authors have addressed most of my comments. Still a minor revision is needed.

Kmers have been widely applied in ML-based sequence-function prediction. Here, I have provided latest references (PMID: 31994694, 31099381, 31157855 and 31542696). It would be appropriate to include it in the following sentence "Topic modelling techniques have been previously used in bioinformatics to classify sequences ether according to their coding"

We reviewed and included the kindly-suggested references as follows:

“Topic modeling techniques have been previously used in bioinformatics to classify sequences ether according to their coding genes [26], 16S rDNA [25], or other biological activities [37–40].”

Round 3

Reviewer 1 Report

arguments accepted.

Reviewer 2 Report

The authors have addressed all my comments. Therefore, I recommend this paper for the publication.